# A Mathematical Model of Value Co-Creation Dynamics Using a Leverage Mechanism

**Tatsunori Hara *** , **Satoko Tsuru and Seiichi Yasui**

Organization for Interdisciplinary Research Projects, The University of Tokyo, Tokyo 113-8656, Japan;
tsuru-satoko@g.ecc.u-tokyo.ac.jp (S.T.); yasui-seiichi@g.ecc.u-tokyo.ac.jp (S.Y.)
* Correspondence: hara-tatsunori@g.ecc.u-tokyo.ac.jp

**Abstract:** Service marketing for sustainability can be addressed in studies on human wellbeing promoted by delight and value co-creation within service systems. However, there is scant research on formal models of value co-creation dynamics with respect to customer delight. This study aimed to formulate a mathematical model of value co-creation dynamics based on a "seesaw catapult" using a lever system. This is based on the concept presented in ISO/TS 24082 and involved service providers' customer centricity and customers' active participation. We solved the maximization problem for a ball's height (i.e., customer experience) by optimizing customers' active participation and the scale of data collection. Theoretical analysis of rotational motion dynamics revealed nonlinear, complementary, and trade-off relationships in the model. Optimal conditions for the variables were obtained, and additional conditions of the service provider's customer centricity were derived to achieve customer delight. In this study, a switchback co-creation process of the service system was constructed based on these findings. This study contributes to the value co-creation literature by providing a normative model of co-creation dynamics that enables deductive development and generates various co-creation processes. Service marketing sustainability can be expanded by exploring appropriate co-creation scenarios that maintain and engage people in service systems.

**Keywords:** co-creation; dynamics of rotational motion; lever system; customer centricity; customer participation; customer delight; service excellence; nonlinearity; trade-off; service engineering

## 1. Introduction

Sustainability, broadly defined, includes the consideration of natural, human, social, and economic capital as its objects. The relationship between service marketing and sustainability is addressed in studies on the following two issues: human-value-oriented service research coupled with the concept of human wellbeing (e.g., [1,2]) and value co-creation within service systems (e.g., [3–5]).

A recent study on transformative service research [2] revealed that customer delight can benefit wellbeing across individual, collective, and societal levels. ISO 23592, "Service excellence—Principles and model", which was published in 2021, conveys the importance of differentiation and a sustainable service business by achieving customer delight [6]. If a customer is delighted, this positively influences their loyalty [7]. Non-financial results based on customer delight include establishing and strengthening long-term customer relationships, long-term cost-saving potential, and improved customer cooperation and engagement [6]. These imply value co-creation [8,9]; however, this is not emphasized in ISO 23592.

The dynamics and complexity of a service system [4,10] may be influenced by the co-creation of value with the customer [3]. Therefore, it is essential to attain a normative understanding of value co-creation dynamics. Several empirical studies have examined the static structure and the linear relationships among the factors involved in value co-creation using structural equation modeling (SEM) (e.g., [11–14]). However, there is not much literature on mathematical models of value co-creation dynamics. Meynhardt et al. introduced

nine systemic principles of value co-creation, such as amplification, nonlinearity and feedback, and phase transitions [5], to explain micro–macro links in service systems. Several multidisciplinary studies have been conducted using multi-agent simulations and game-theoretic analyses relevant to this systemic property of value co-creation (e.g., [15–18]), but a unified co-creation process between the service provider and the customer [19] has not been presented. Durugbo and Pawar formalized a unified model that characterized the co-creation process [19]. However, it focused on a descriptive model and, thus, did not provide normative and prescriptive models of the co-creation dynamics.

Hara et al. constructed a qualitative dynamic model of co-creation for the knowledge-based transition to a provider–customer relationship in servitization [20]. It specified interconnections among activity cycles of providers and customers that resulted in the transformation of service provision. Ho and Shirahada studied mechanisms for maintaining service systems and proposed "service mechanics", representing dynamic mechanisms that affect actor transformation in co-creation [21]. However, mathematical models that elaborate on such co-creation processes are necessary for further developments.

ISO/TS 24082 "Service excellence—Designing excellent service to achieve outstanding customer experiences" differentiates itself from general service design by focusing more on customer delight and co-creation with customers [22]. It also recommends designing a co-creation environment [23,24] between customers and service providers as a leverage mechanism in order to enhance the customer experience and better sustain customer delight. The conceptual effect of the leverage mechanism was presented in [22,24] as a "seesaw catapult". The greater the maximum height of the launched ball, the better the customer experience. However, a theoretical analysis was not presented.

This study aims to develop a mathematical model of value co-creation dynamics using this leverage mechanism. The "seesaw catapult" is a metaphor [25] for value co-creation dynamics. The formal model contributes to the value co-creation literature, since it enables deductive reasoning of co-creation processes based on the dynamics. Compared to generic co-creation dynamics [26,27] that are patterns or processes of change, growth, or activity, the dynamics used in this study follow classical mechanics in physics. Thus, the study of value co-creation dynamics represents and theorizes forces and their relation primarily to the motion relevant to co-creation that focuses on value acquisition. We formulate the dynamics of rotational motion and optimize the objective function for the ball's maximum height. Nonlinearity, trade-off relationships, and the required balance are considered in the analysis. Based on the findings, a switchback co-creation process of the service system for customer delight is constructed. Thus, this study contributes to service marketing sustainability by exploring systemic value co-creation and its processes for delight, which helps actors maintain the service and engage in practices of wellbeing.

## 2. Materials and Methods

### 2.1. Basic Idea of the Seesaw Catapult System

#### 2.1.1. Service Provider's Customer Centricity and Customer's Active Participation

A co-creation environment supports intense cooperation between service providers and customers at touchpoints. The level of intense cooperation is determined by the service provider's customer centricity and the customer's active participation. Figure 1 shows these relationships based on the value creation sphere [8]. Here, customer centricity is defined as the customer orientation, which focuses on value creation and value acquisition [22]. The service provider's customer centricity covers the provider sphere and the joint sphere. The customer's active participation is demonstrated via various customer behaviors directed toward the service provider's organization in both the joint and customer spheres. The customer is the value creator in direct interaction, but the value may be co-created with the provider [8]. This depends on the willingness of customers to fulfill their role, e.g., customer role readiness [23]. This study focuses on the joint sphere in which the process of intense cooperation emerges.

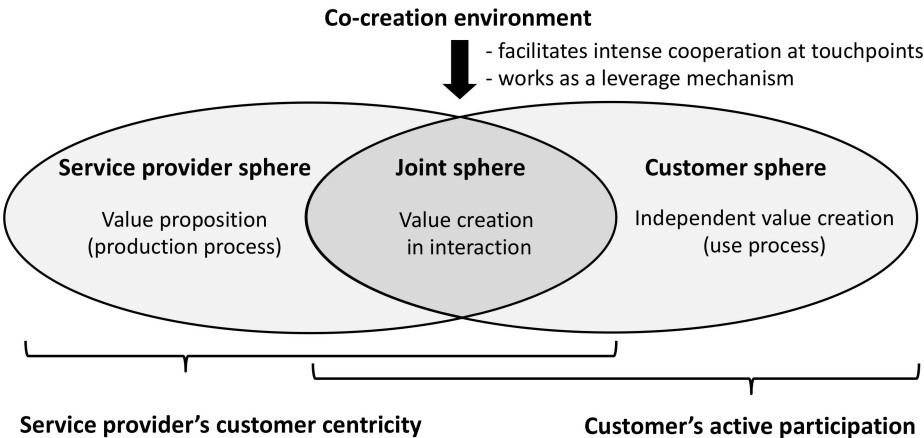

**Figure 1.** Value creation sphere and the elements considered in this study.

### 2.1.2. Rotational Force of the Lever System

Hara et al. originally illustrated the strength of the intense cooperation between the customer and service provider as the area of a parallelogram using two vectors [24], as shown in Figure 2a. The area is calculated using the cross-product of the two vectors. Subsequently, the analogy of a lever system was introduced; its rotational force (the moment of force in physics) is based on the cross product.

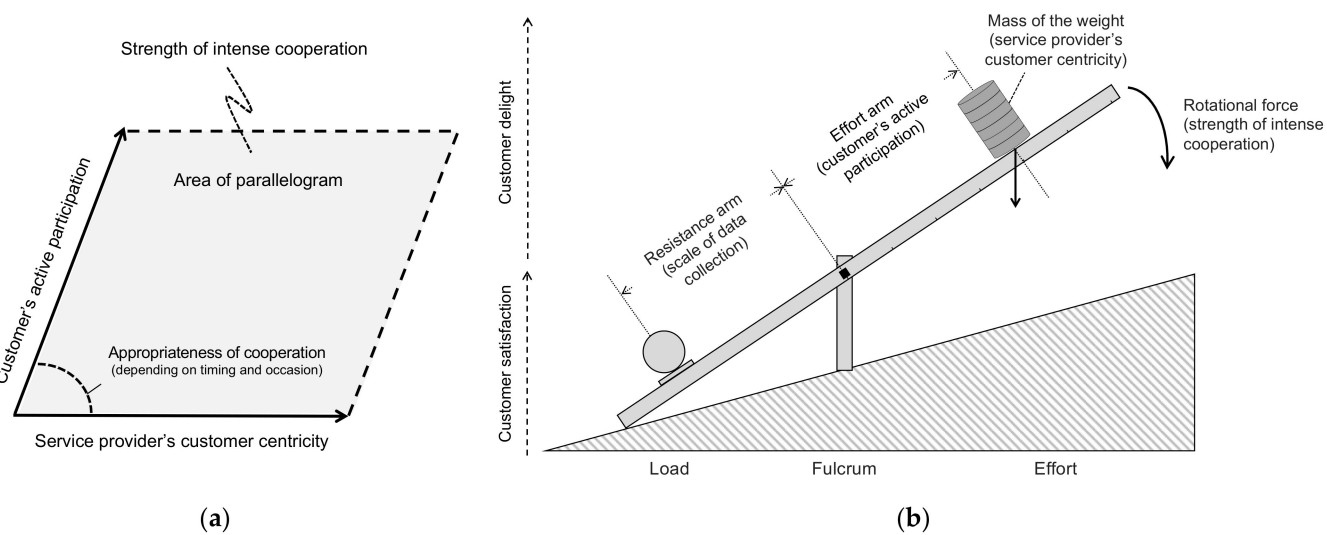

(**a**)                                                                 (**b**)

**Figure 2.** Schematic illustrations of leverage mechanisms for co-creation. (**a**) Area of the parallelogram. (**b**) Rotational force of the lever system (adapted with permission from ISO/TS 24082:2021, Figure D.1 in Ref. [22]. Copyright 2021, ISO).

Figure 2b shows a seesaw catapult of the ball using the lever system. The moment of force rotates the platform down (i.e., a clockwise rotational force). The effort owing to the weights represents the service provider's customer centricity, the current level of which corresponds to the weight. The effort arm's length (the distance between the fulcrum and the effort) represents the customer's active participation. Its current level determines the position at which the weights should be placed. The greater the effort and the effort arm, the greater the clockwise rotational force. Different levels of the customer centricity of service providers and the active participation of customers can be considered. Examples are listed in Table A1.

Using this structure, ISO/TS 24082 describes a basic service for customer satisfaction. In the case of less intense cooperation, the platform rotates slowly, and the ball does not

move upward because of its reduced momentum. High customer satisfaction can be achieved, but customer delight is not expected.

In contrast, excellent service based on co-creation causes the platform to launch the ball upward with more intense cooperation. The ball's maximum vertical distance from the platform in the horizontal plane is indicative of an outstanding customer experience. The greater the ball's speed during release from the platform is, the more likely the production of outstanding customer experiences via co-creation will be.

However, not every ball jump can create customer delight. This study investigates the conditions that result in customer delight.

### 2.1.3. Trade-Off Relationship and the Scale of Data Collection

It is first necessary to understand the concept of a rotational force, but if we carefully consider the mechanics [28–30], trade-off relationships are observed, especially on the left side, as follows: "amplification of the ball speed" and "difficulty in rotation of the platform owing to the ball's mass and position". These complicate the behavior of this mechanism, but also hint at the importance of balance in service.

The former indicates that a ball placed further from the fulcrum will have a greater speed when released (which is called the "catapult"). In the latter, the ball's mass generates the counterclockwise rotational force and rotational inertia. Therefore, the longer the platform and the heavier the load, the more difficult it is for the platform system integrated with the ball to rotate around the fulcrum.

Hence, there is no simple solution for the resistance arm length, i.e., the distance between the fulcrum and the ball. In Figure 2b, the resistance arm is associated with the scale of data collection, which contributes to the co-creation environment. In recent years, data collection in the service delivery process has become crucial so that service providers can utilize the data in ways including feedback provision, service personalization, and learning. The amplification can be the effect of data utilization, and the difficulty can be the cost of data collection and utilization. The most appropriate scale of data collection to obtain the greatest speed of the ball is determined based on the remainder of the system and is not too small or too large.

### 2.2. Dynamics of Rotational Motion

This study formulates the mechanism discussed in Section 2.1 according to the variable notations given in Figure 3. The mechanism is modeled as a rigid body rotation [28–30]. The moment of force $M$ around the fulcrum is given by

$$M = m_l g L_l \cos \theta - m_r g L_r \cos \theta, \tag{1}$$

where $\theta$ is the angle measured from the horizontal plane. Note that $\theta$ is positive in the counterclockwise direction and decreases with clockwise rotational motion.

As mentioned in Section 2.1.3, the rotational motion of a rigid body needs to consider the difficulty in rotation caused by rotational inertia. Neglecting the platform's mass, the rotational inertia $I$ (the moment of inertia in physics) of this system is given by

$$I = m_l L_l{}^2 + m_r L_r{}^2. \tag{2}$$

Then, the angular acceleration $\ddot{\theta}$ is given by

$$\ddot{\theta} = \frac{M}{I} = \frac{m_l L_l - m_r L_r}{m_l L_l{}^2 + m_r L_r{}^2} g \cos \theta. \tag{3}$$

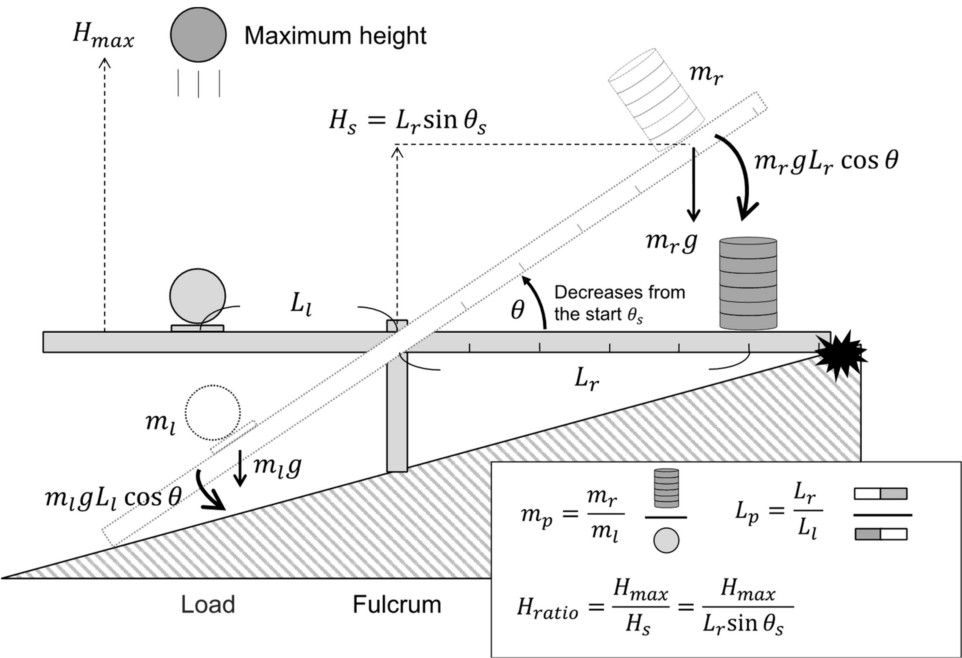

$m_r$:    Mass of the weight (service provider's customer centricity)
$L_r$:    Effort arm's length (customer's active participation)
$L_l$:    Resistance arm's length (the scale of data collection)
$m_l$:    Mass of the ball (organizational inertia)
$H_{max}$:  Maximum height of the ball (the degree of customer experience)
$\theta$:    Angle of the platform relative to the horizontal
$\theta_s$:    Initial angle
$H_s$:    Starting height of the weight
$g$:    Gravitational acceleration (constant)
$m_p$:    The ratio of the service provider's customer centricity to organizational inertia
$L_p$:    The ratio of the customer's active participation to the scale of data collection

**Figure 3.** Variable definitions of the seesaw catapult system analyzed.

*2.3. Ball's Maximum Height: Objective Function*

Equation (3) is a second-order nonlinear ordinary differential equation. Solutions of $\theta$ and $\ddot{\theta}$ at a given time are generally obtained through an analysis using Jacobi's elliptic function or a numerical analysis based on the Runge–Kutta method [28]. However, the process of rotational motion is not considered in this analysis, but, rather, the result when the ball is released from the platform. The conservation of energy between the initial state and the horizontal state is given by

$$\frac{1}{2}I\omega_h^2 = m_r g L_r \sin \theta_s - m_l g L_l \sin \theta_s, \tag{4}$$

where $\theta_s$ is the initial angle, $\omega_h$ is the angular acceleration in the horizontal state, and $m_r L_r > m_l L_l$ is assumed to consider only cases for which the lever system is driven.

Let $L_p$ be the arm ratio $L_r/L_l$ and let $m_p$ be the mass ratio $m_r/m_l$ (both dimensionless). $L_p$ and $m_p$ both represent the ratio of right to left. Then, we have:

$$\omega_h^2 = \frac{2(m_p L_p - 1)}{L_l(1 + m_p L_p^2)} g \sin \theta_s. \tag{5}$$

The velocity of the ball on the platform is given by $v = L_l \omega$. Using the speed $v_h$ at release, the ball's maximum height is given by

$$H_{max} = \frac{v_h^2}{2g} = \frac{(L_l \omega_h)^2}{2g} = \frac{(m_p L_p - 1)}{1 + m_p L_p^2} L_l \sin \theta_s. \tag{6}$$

Given that each possible design variable is involved in both the denominator and the numerator of Equations (3) and (6), nonlinearity arises in this mechanism and the ball's maximum height.

In this investigation, $m_l$, the organizational inertia, is not controlled. Thus, $m_p$ is assumed to be proportional to $m_r$, the service provider's customer centricity.

*2.4. Design Variables and Delight Criterion*

In this study, the maximization problem is solved, where the objective function is $H_{max}$. Since its first-order derivative with respect to $m_p$ is always positive, $H_{max}$ is a strictly increasing function of $m_p$. Sections 3.1 and 3.2 consider $L_l$ and $L_r$, respectively, as design variables. We will obtain the conditions for $L_l$ and $L_r$ when $H_{max}$ is maximized.

$H_{max}$ depends on the balance of $m_p$, $L_r$, and $L_l$, and it is proportional to them. Therefore, we introduce the start height of the weight $H_s$ as a criterion for evaluating $H_{max}$ in terms of the co-creation effect. This is referred to as the "delight criterion". If $H_{max}$ exceeds this criterion, we assume that delight is created through co-creation. This delight criterion corresponds to the "desired service level" in the theory of customer expectation and the zone of tolerance [31,32]. Exceeding the zone of tolerance can surprise and delight customers.

An example of the balanced state of the lever system is given by $(m_p, L_r, L_l) = (1, 1, 1)$; the same conditions exist on both sides. However, if $(m_p, L_r, L_l) = (3, 1, 1)$, we obtain $H_{max} : H_s = 1 : 2$. Therefore, this gives a basic understanding of the effect of the catapult system as: "When the mass ratio is tripled relative to the balanced state, the ball jumps up to half the start height of the weight".

## 3. Results

*3.1. Case 1: Changing the Design Variable "Customer's Active Participation"*

3.1.1. Maximizing the Ball's Height

Let $L_l$ be fixed and let $L_r$ be the design variable of $H_{max}$. We solve the maximization problem and obtain the conditions of $L_r$ when $H_{max}$ is maximized.

First, Equation (6) is treated as a function of $L_p$. Then, $L_p^*$, the solution of $L_p$ when $H_{max}$ assumes the global maximum, is given by:

$$L_p^* = \frac{1 + \sqrt{1 + m_p}}{m_p}. \tag{7}$$

An alternative condition of $L_r$ represented by $L_l$ and $m_p$ is

$$L_r^* = \frac{1 + \sqrt{1 + m_p}}{m_p} L_l. \tag{8}$$

It is possible to determine the most appropriate level of "active customer participation" that maximizes customer experience based on the given variables of "service provider's customer centricity" and "the scale of data collection". We consider that a better customer experience is provided under this condition. Compared to the balanced state ($L_r = L_l/m_p$), $L_r^*$ is increased by a factor of $\sqrt{1 + m_p}/m_p$. The larger the value of $m_p$, the smaller the necessary surplus. For example, the optimal conditions were $L_r^* = 1.72 L_l$ if $m_p = 1.5$ and $L_r^* = 1.36 L_l$ if $m_p = 2$.

The optimal $H_{max}$ is obtained in a simplified form. If $L_p = L_p^*$, then, we have:

$$H_{max}\left(L_p = L_p^*\right) = \frac{1}{2}L_l \sin \theta_s \left(\sqrt{1 + m_p} - 1\right). \tag{9}$$

The maximum of $H_{max}$ is proportional to the square root of $m_p$. Figure 4 shows the function of $H_{max}$ of $L_p$ if $m_p = 4$. Note that $L_l$ is fixed.

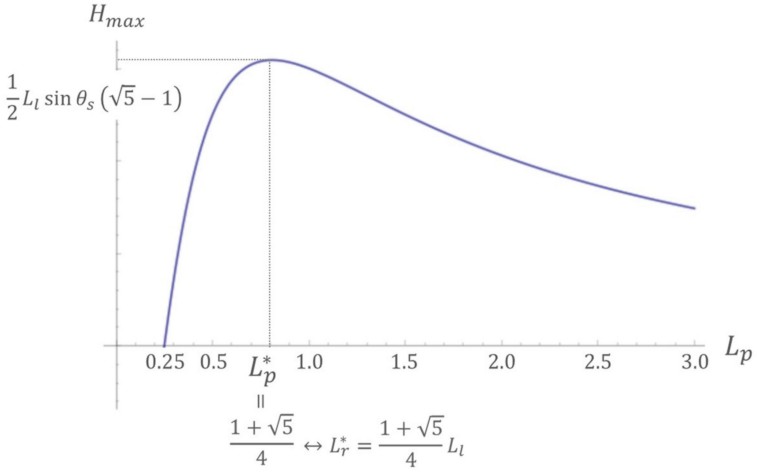

**Figure 4.** Function $H_{max}$ ($m_p = 4$) of $L_p$ and the optimal customer active participation ($L_r^*$).

### 3.1.2. Additional Delight Condition

Here, we assume that the optimal conditions (Equations (7) and (8)) hold and consider an additional condition in which $H_{max}$ exceeds the start height of the weight $H_s$, i.e.,

$$H_{ratio}\left(L_p^*\right) = \frac{H_{max}\left(L_p^*\right)}{H_s} = \frac{1}{2}\frac{1}{L_p}\left(\sqrt{1 + m_p} - 1\right) \geq 1. \tag{10}$$

As a result, the additional condition of $m_p$ that leads to delight in the maximized customer experience is given by

$$m_p \geq 2\left(1 + \sqrt{2}\right) \approx 4.83. \tag{11}$$

The achievement of delight via the design variable optimization of $L_r$ depends only on $m_p$, not $\theta$ or $L_l$. For simplification, this condition is dealt with as $m_p \geq 5$ in this study.

### 3.1.3. Process of Service Development Leading to Customer Delight

The following service development process can be assumed based on the previous analysis.

1.  An organization X had insufficient organizational capability, indicating reduced service provider's customer centricity. Thus, we supposed that $m_p < 5$.
2.  The organization strived to enhance the customer experience ($H_{max}$), even under the current condition of $m_p < 5$, by encouraging more appropriate active participation of the customer. However, this did not achieve delight, which satisfied Equation (10).
3.  The organization worked on improving customer centricity, and $m_p \geq 5$ was realized. Accordingly, the most appropriate level of active participation was recalculated and pursued by encouraging and moderating its current level.

As a result, the possibility of customer delight via co-creation increased.

*3.2. Case 2: Changing the Design Variable "the Scale of Data Collection"*

3.2.1. Maximizing the Ball's Height

Let $L_r$ be fixed and let $L_l$ be the design variable in $H_{max}$. We solve the maximization problem and obtain the conditions of $L_l$ when $H_{max}$ is maximized.

An alternative form of Equation (6) that is obtained when $L'_p = 1/L_p = L_l/L_r$ is given as:

$$H_{max}\left(L'_p\right) = L_r \sin\theta_s \frac{L'^2_p\left(m_p - L'_p\right)}{m_p + L'^2_p}. \tag{12}$$

$L'^*_p$, the solution of $L_p$ when $H_{max}$ is the global maximum, is given as

$$L'^*_p = \sqrt[3]{\sqrt{m_p{}^4 + m_p{}^3} + m_p{}^2} - \frac{m_p}{\sqrt[3]{\sqrt{m_p{}^4 + m_p{}^3} + m_p{}^2}}, \tag{13}$$

after solving a cubic equation, subject to $L_p > 0$.

This $L'^*_p$ is complicated, but can be linearly approximated. An example of the approximation formula is given as follows:

$$L'^*_p = \frac{1}{2}m_p. \tag{14}$$

This implies that $L'^*_p < m_p$. Note that this approximate formula is not unique and was obtained using data intervals of $1 \leq m_p \leq 7$ and a discretization of 0.1. The coefficient of determination $R^2$ was 0.997.

An alternative condition for $L_l$ using $L_r$ and $m_p$ is given as:

$$L^*_l = \frac{1}{2}m_p L_r. \tag{15}$$

It is possible to determine the most appropriate level of "the scale of data collection" that maximizes customer experience based on the other variables of "service provider's customer centricity" and "customer's active participation". We consider that a better customer experience is provided under this condition. A better strategy would be to reduce the length of the left arm (the scale of data collection) by half relative to the balanced state ($L_l = m_p L_r$).

The optimal $H_{max}$ is obtained in a simplified form if $L'_p = L'^*_p$ and is given by,

$$H_{max}\left(L'_p = L'^*_p\right) = \frac{1}{2}L_r \sin\theta_s \frac{m_p^2}{m_p + 4}. \tag{16}$$

Figure 5 shows the function $H_{max}$ of $L'_p$ if $m_p = 3$. Note that $L_r$ is fixed. The value of $L'^*_p$, 1.5, is plotted as the approximate solution based on Equation (15). Its exact solution is 1.57.

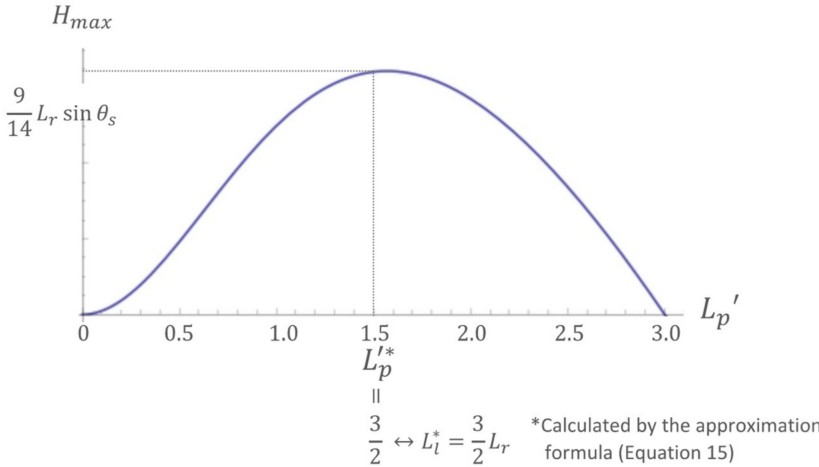

**Figure 5.** Function $H_{max}$ ($m_p = 3$) of $L'_p$ and the optimal scale for data collection ($L^*_l$).

### 3.2.2. Additional Delight Condition

Here, we assume that the optimal condition (Equations (14) and (15)) holds and consider an additional condition in which $H_{max}$ exceeds the start height of the weight $H_s$, i.e.,

$$H_{ratio}\left(L'^*_p\right) = \frac{H_{max}\left(L'^*_p\right)}{H_s} = \frac{m_p^2}{2(m_p + 4)} \geq 1. \tag{17}$$

As a result, the additional condition of $m_p$ that leads to delight when the customer experience is maximized is given by:

$$m_p \geq 4. \tag{18}$$

We adopt this condition for simplification, even though it varies depending on the approximation formula, such as Equation (14).

### 3.2.3. Process of Service Development Leading to Customer Delight

The following service development process can be assumed based on the preceding analysis.

1.  An organization X had insufficient organizational capability, resulting in reduced customer centricity of service providers. Thus, we suppose that $m_p < 4$ holds.
2.  The organization strived to enhance the customer experience ($H_{max}$), even under the current condition of $m_p < 4$, by encouraging a more appropriate scale of data collection. However, delight was not achieved, which satisfied Equation (17).
3.  The organization worked on improving customer centricity, and $m_p \geq 4$ was realized. Accordingly, the most appropriate scale for data collection was updated and pursued by expanding and reducing its current level. As a result, the possibility of customer delight via co-creation increased.

### 3.3. Switchback Co-Creation Process for Customer Delight

In the previous two cases, either the scale of data collection ($L_l$) or the customer's active participation ($L_r$) was fixed, and the other design variables were optimized. In practice, instead of always fixing one variable, it is worth investigating a method that gradually improves customer experience by alternately optimizing the design variables $L_l$ and $L_r$. Furthermore, as described in Sections 3.1.3 and 3.2.3, it is necessary to increase $m_p$ at a certain stage to achieve delight.

This study proposes a switchback co-creation process for customer delight, and Figure 6 shows a result for an exemplified case. This case is comprised of states 0 to 11. The horizontal axis represents $L_p$ and the vertical axis represents $H_{max}$. The bubble size

of the nodes represents the values of $m_p$. The state 0 occurs for $(m_p, L_r, L_l) = (3, 1, 1)$, which was a reference state of the catapult system in Section 2.4.

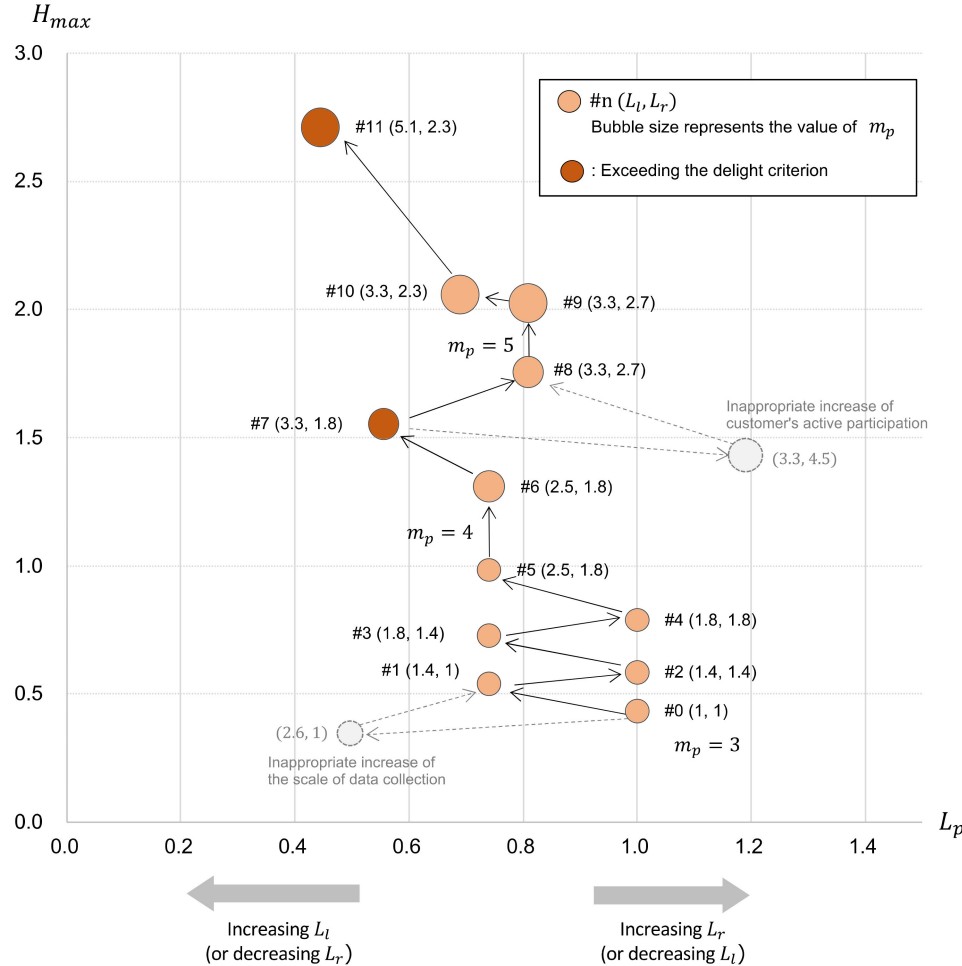

**Figure 6.** Numerical analysis of the switchback co-creation process for customer delight.

The figure shows how the customer experience ($H_{max}$) is enhanced by alternately optimizing the changing directions of the "customer's active participation" and "the scale of data collection", similarly to a switchback system. Considering the overall increase in data collection and utilization, the active participation of customers can be encouraged. As customer active participation increases, data collection can be enhanced to capture the emotional responses of customers. This process supports sustainable development. In state 7, the delight criterion is exceeded for the first time after $m_p$ evolves to 4. However, once achieved, the objective function is still $H_{max}$. As in the subsequent state 8, it may fall below the delight criterion, even though the customer experience is improved. In this example, the delight criterion is again exceeded in state 11.

The customer's active participation is moderated in the transition from state 9 to state 10. The evolution of $m_p$ to 5 in the previous operation primarily impacts the customer experience. As customer orientation is enhanced, extra "customer's active participation" is appropriated.

Figure 6 also shows two possible imbalanced states during the co-creation process: an excessive scale of data collection and an excessive customer active participation. Unfortunately, these are not commensurate with the other variables in each case, so the customer experience is adversely impacted.

## 4. Discussion

In this study, the dynamics of co-creation were formulated and analyzed based on the "seesaw catapult" using a lever system. Trade-off relationships and the required balance among variables on both sides of the system were analyzed. Using the ball's maximum height as an objective function, we demonstrated two cases of design variable optimization: designing the customer's active participation and designing the scale of data collection. Table 1 summarizes the results for each case.

**Table 1.** Results for the analysis of value co-creation dynamics.

| | Case 1: Designing the Customer's Active Participation | Case 2: Designing the Scale of Data Collection |
|---|---|---|
| $H_{max}$ | | $\frac{(m_p L_p - 1)}{1 + m_p L_p{}^2} L_l \sin \theta_s$ |
| Maximum of $H_{max}$ | $\frac{1}{2} L_l \sin \theta_s \left( \sqrt{1 + m_p} - 1 \right)$ | $\frac{1}{2} L_r \sin \theta_s \frac{m_p^2}{m_p + 4}$ |
| Optimal design variable | $L_r = \frac{1 + \sqrt{1 + m_p}}{m_p} L_l$ | $L_l = \frac{1}{2} m_p L_r$ |
| Additional condition to exceed the delight criterion | $m_p \geq 5$ | $m_p \geq 4$ |

In Case 1, the larger the value of $m_p$, the smaller the necessary surplus of $L_r$ compared to the balanced state ($L_r = L_l / m_p$). This indicates that the improvement of the customer centricity of the service provider complements customer participation. Excessive customer active participation may cause difficulty in rotation, so this should be managed to meet the current service provider's capability.

The result of Case 2 was simpler compared to that of Case 1. This half-length strategy relative to the balanced state also implies that the most appropriate $L_l$ is still proportional to the given $m_p$ and $L_r$. This strategy balances the amplification of the ball speed as the effect of data utilization and the difficulty in rotation as the cost of data collection and utilization.

Additional conditions were obtained for creating delight, including $m_p \geq 5$ and $m_p \geq 4$. The demand for $m_p$ is higher in Case 1 because the customer's active participation affects the start height of the weight. The realization of delight by exploiting high customer participation requires greater customer centricity of the service provider, which constitutes $m_p$. Customers who are actively involved in the service may have high expectations, as explained in Section 2.4.

This paper assumed that escalating $m_p$ in the switchback co-creation process is performed by improving the service provider's customer centricity $m_r$. However, the variable $m_l$, the ball's mass as organizational inertia, may also be considered. Organizational inertia represents the degree to which organizations continue to operate in the usual way instead of responding to environmental changes [33]. We assume that the smaller the organizational inertia is, the greater the number of organizational activities related to service excellence performed in practice will be. Additional delight conditions of $m_p$ imply that there exists a need to improve the comprehensive organizational capability at crucial points. ISO 23592 outlines recommendations of activities to improve such organizational capability, including management and cultural aspects. Therefore, the elucidation of levels of $m_l$ based on the contents of ISO 23592 will show how service excellence achieves customer delight through co-creation.

The implications of this study for researchers and practitioners are explored in the next section.

## 5. Implications

### 5.1. Theoretical Implications

The formalization of the unified co-creation process by Durugbo and Pawar [19] was descriptive and suitable for analyzing existing scenarios. However, it was not easy to clarify the generic dynamics of co-creation and prescribe various patterns of co-creation

processes. This study devised a deductive model based on an analogy with physical behavior. The "seesaw catapult" metaphor shapes our perception and understanding of value co-creation dynamics. Deductive research starts with an extant theory, forcing reality into its format [34]. It enables the logical deployment of a system based on the premise. Furthermore, the proposed mathematical model and formal approach transcend the analogy and offer higher levels of abstraction. Thus, this study contributes to the value co-creation literature by providing a normative model and a deductive development of co-creation dynamics.

The presented switchback co-creation process is one of the results of the development. It prescribes the evolution of the service system toward the provider–customer relationship [20]. Rather than specializing in any one variable, multiple variables can be improved in stages, depending on the situation at a given time, so that the service system can evolve in sustainable ways. Optimization during gradual improvement may suggest a reduction of excessive actions. Value can be co-created by seeking and understanding nonlinear dynamics in a service ecosystem [5]. This study successfully dealt with nonlinear, complementary, and trade-off relationships in the optimization process. These relationships have not been incorporated in previous investigations of value co-creation.

Further discussions on customer delight based on the model could also contribute to knowledge on the zone of tolerance [31,32] and the service excellence pyramid [6,35], which explains the structure of excellent service. Customer delight can benefit aspects of wellbeing addressed in transformative service research [2] and can help actors engage in practices of wellbeing. In addition, the evolution of a customer's active participation leading up to delight may include design aspects, such as so-called participatory design [36], user design, or design-in-use [37]. A technical tool for the design-in-use activities of customers is helpful for advancing continuous co-creation [20], and its feedback enables service providers to gain customer insights. By including these ideas, service marketing sustainability can be explored by configuring appropriate processes to maintain and engage actors in each service system.

This study may also encourage interdisciplinary research with engineering studies. For example, in the early 2000s, Arai and Shimomura proposed a service modeling method based on the engineering discipline of conceptual design, starting with a definition of service [38]. The method was developed for service engineering in a top-down manner, and a piece of computer-aided design software called service CAD was implemented while incorporating the conventional service marketing literature. As with the software and its series of studies (e.g., [39,40]), the presented mathematical model contributes to developing new service engineering based on the co-creation concept, which has been preliminarily developed in service marketing.

*5.2. Practical Implications*

For service firms, the presented model can be used as an analytical tool to review the current balance of each service. Furthermore, they can configure and explore various co-creation processes using operations on the model according to the situation of each service. The switchback process is a normative example. For instance, customer participation is essential in co-creation, but increasing it is more constrained than it is with other factors in practical cases. For vulnerable customers [21], the ideal level of participation calculated may not be realistic. In such cases, it is preferable to improve other factors gradually, including, primarily, $m_p$. If they are sufficiently improved, a targeted equivalent level of customer experience can be achieved and maintained with a lesser increase in participation. Then, a mechanism of actor transformation in co-creation [21] that intensifies the customer's active participation will be considered.

A new practical specification, ISO/TS 23686, regarding the measurement of service excellence performance will be published in 2022 [41]. This specification recommends measurement methods and metrics for organizational capability, employee engagement, customer experience, and customer delight. Service firms could identify the levels of

variables used in the proposed model by aggregating the measured performance based on the specification. However, the levels of variables listed in Table A1 can be improved. The result of the numerical analysis in the switchback co-creation process facilitates the development of a set of new levels for operationalizing variables in terms of co-creation.

The dynamics of co-creation are difficult to experience because they involve temporal variation and nonlinearity. In physics education, workshops on catapults are occasionally introduced for effective learning and as a motivational tool [30]. The beer game on the logistical system [42] is an introductory exercise in system thinking to experience significant system behaviors, such as time delays and bullwhip effects. If we advance this research and implement it as a computer simulation and a tangible device, educational use for value co-creation dynamics is expected.

## 6. Limitation

Even though the theoretical model of this study may allow for a normative understanding of value co-creation, this understanding and the main findings rely on the initial premises in the model being correct and may not be directly applied to the actual service fields.

Optimal design variables are obtained as continuous values; it may not be easy to interpret and deal with them in practice for the discrete levels, as shown in Table A1, even if their neighborhood value is acceptable. Regarding the scale of data collection, their levels for reference are not yet compiled as they are in the other variables. Examples of levels that consider both benefits and the cost of data collection and utilization would facilitate a better understanding of the behavior of the developed model.

## 7. Conclusions

This study developed a mathematical model of value co-creation dynamics based on the seesaw catapult using a lever system. Using the ball's maximum height as an objective function, we demonstrated two cases of design variable optimization and the switchback co-creation process. This study successfully dealt with nonlinear, complementary, and trade-off relationships in the dynamics. Additional conditions of the service provider's customer centricity were derived to achieve customer delight. This study contributes to the value co-creation literature by providing a normative model of co-creation dynamics that enables deductive development and generates various co-creation processes. It also contributes to service marketing for sustainability by exploring appropriate co-creation scenarios that maintain and engage people in each service.

Future work will include the capture of the temporal process of rotational motion in detail, which calls for an interpretation of the effects of the angles of the platform relative to the horizontal $\theta$ and $\theta_s$.

**Author Contributions:** Conceptualization, T.H., S.T. and S.Y.; methodology, T.H.; software, T.H.; validation, T.H. and S.T.; formal analysis, T.H. and S.Y.; investigation, T.H.; resources, T.H., S.T. and S.Y.; data curation, T.H.; writing—original draft preparation, T.H.; writing—review and editing, S.T. and S.Y.; visualization, T.H.; project administration, S.T.; funding acquisition, S.T. All authors have read and agreed to the published version of the manuscript.

**Funding:** This research received no external funding.

**Institutional Review Board Statement:** Not applicable.

**Informed Consent Statement:** Not applicable.

**Data Availability Statement:** Not applicable.

**Acknowledgments:** The authors gratefully thank all experts and contributors in ISO/TC 312/WG 2 for creating ISO/TS 24082, which provides the research background of this study.

**Conflicts of Interest:** The authors declare no conflict of interest.

## Appendix A

The different levels of service provider's customer centricity and customer's active participation should be understood. Examples are given in Table A1. The service provider column lists the rationale for their behavior corresponding to each level of customer centricity.

**Table A1.** Examples of levels of service provider's customer centricity and customer's active participation (adapted with permission from ISO/TS 24082:2021, Table C.1 and Table C.2 in Ref. [22]. Copyright 2021, ISO).

| Level | Service Provider's Customer Centricity | Customer's Active Participation |
|:---:|:---:|:---:|
| 1 | Rewards | Acceptance |
| 2 | Regulations | Express needs clearly |
| 3 | Requests from customers | Use efficiently and effectively |
| 4 | Observation from the customer's point of view | Provide feedback |
| 5 | Empathy for customers | Recommend to others |
| 6 | Social interest (or community feeling) | Feel psychological ownership [43] |

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
