# Peer review of "A Mathematical Model of Value Co-Creation Dynamics Using a Leverage Mechanism"

_sustainability, doi:10.3390/su14116531_

Round 1

Reviewer 1 Report

Dear Authors,

I find your manuscript interesting and well written. 

However, there are some minor issues:

First, discussion is not similar with the paper's implications or contributions. This must be a separate section.

Second, I do not see the point in having the Theoretical implication section divided in three separate subsections.

Third, reorganize the paper to have all the implications in the same section.

Fourth, I find the practical implications very weak. This sectin needs to be extended because it is of special interest for readers.

Finally, some sort of proofreading is needed because there are many typos.

Reviewer 2 Report

The presentation reflects the present state of knowledge. The paper is very well structured. The Introduction section is good, in this section the authors presents clearly the objectives and the main contributions of the study. The authors had provided sufficient background and include relevant references. The method is adequately described. The results are clearly presented. The conclusions are supported by the results. 

Author Response

Thanks for assessing the manuscript. The conclusion section was added and made clear according to other reviewers' comments.

Reviewer 3 Report

Dear author(s),

Thank you for the interesting read. It is a good work. However, you should have a proper conclusion and separate literature review section.

Good luck.

Author Response

The conclusion section was added with future works. However, the literature review was kept in the introduction section. The authors appreciate the suggestion, but it was difficult to separate (and expand) the literature review with the current amount and within the minor revision. Instead, we moved the explanations related to Figure 1 from the introduction section to Section 2.1 so that readers reach the objective of this study earlier. The authors hope this meets the suggestion.

Reviewer 4 Report

This paper has been well written, but needs further justifications.

  1. In the introduction, the authors should explain key contributions of why readers have to read this article.  Theoretical contributions are essential.
  2. While the current study focused on the mathematical approach of value co-creation, the authors must explain why these promotions models are valuable.
  3. Finally, in the discussion, both theoretical and practical implications should be improved. In addition, the authors must add a new section of Conclusion.

Good luck!
